

# Comparison of RO tropopause height based on different tropopause determination methods

Ziyan Liu[1,2,3], Weihua Bai[1,2*], Yueqiang Sun[1,2*], Junming Xia[1,2], Guangyuan Tan[1,2,3], Cheng Cheng[4], Qifei Du[1,2], Xianyi Wang[1,2], Danyang Zhao[1,2], Yusen Tian[1,2,3], Xiangguang Meng[1,2], Congliang Liu[1,2], Yuerong Cai[1,2], Dongwei Wang[1,2]

[1]National Space Science Centre, Chinese Academy of Sciences (NSSC/CAS), Beijing 100190, China;
[2]Beijing Key Laboratory of Space Environment Exploration, Beijing 100190, China
[3]University of Chinese Academy of Sciences, Beijing 100049, China
[4]State Intellectual Property Office of the P.R.C, Beijing 100088, China

*Correspondence to*: Weihua Bai (baiweihua@nssc.ac.cn) and Yueqiang Sun (SYQ@nssc.ac.cn)

**Abstract.** Tropopause region is a significant layer among the earth's atmosphere, receiving increasing attention from atmosphere and climate researchers. To monitor global tropopause via radio occultation (RO) data, there are mainly two methods, one is the widely used temperature lapse rate method, and the other is bending angle covariance transform method. In this paper, we use FengYun3-C (FY3C) and Meteorological Operational Satellite Program (MetOp) RO data and European Centre for Medium-Range Weather Forecasts (ECMWF) reanalysis data to analyse the difference of RO tropopause height calculated by the two methods mentioned above. To give an objective and complete analysis, we first take ECMWF lapse rate tropopause (LRT) height (LRTH) as reference to discuss the absolute bias of RO LRTH and RO bending angle tropopause (BAT) height (BATH), and then give the comparison results between RO LRTH and corresponding RO BATH as supplement to analyse the difference between tropopause height derived from the above two methods. The results indicate that BATH show consistent 0.8-1.2 km positive bias over tropics and high latitude region compared with LRTH, and over mid latitude region, results of BATH show less stability. Besides, the mean bias between BATH and LRTH presents different symmetrical characteristic during 2017.12-2018.2 (DJF) and 2018.6-2018.8 (JJA). However, the mean value of both LRTH and BATH show the similar tropopause variation trend, indicating the availability of both two methods.

**Key word:** Validation, FY3C, MetOp, radio occultation, tropopause

## 1 Introduction

Tropopause region is an important layer of atmosphere structure, which have drawn wide attention in atmospheric research field. The exchange of energy, air mass and water vaper take place across the tropopause (Holton et. al. 1995), which is closely associated to the deep convection and the Brewer-Dobson circulation (Randel et. al. 2006). Also, the variation of tropopause structure leads to the change of stratospheric moisture and the stratospheric chemistry (Randel et. al. 2004; Fueglistaler et. al. 2009). Besides, the variation trend of tropopause is also an indicator of the effect of anthropogenic activities on the environment



(Santer et. al. 2003; Pielke 2004). For example, the uptrend of global tropopause height has strong correlation with the emission of carbon-dioxide as well as the depletion of stratospheric ozone content (Lorenz et. al 2007; Steinbrecht et. al. 1998).

Traditional methods for sounding tropopause structure are based on direct sounding technic or model data, such as radiosonde, and reanalysis data. However, the direct sounding technic usually has an uneven global distribution, with only sparse data
distribution over the south hemisphere. Also, the variable quality of different direct sounding technic, including the different observing method, resolution and data type, makes it hard for data collation and thus severely limits the utility of this kind of data for detecting climate change. The reanalysis data can provide global and temporal coverage and uniform data type, but it cannot be used in real-time observation for it is based on model-driven (Sturaro et. al. 2003; Sterl et. al. 2004), and sometimes reanalysis data suffers from the coarse vertical resolution, which may lead to the missing of tropopause character (Birner et.
al. 2006).

Global navigation satellite system (GNSS) radio occultation (RO) technic is a limb-sounding remote sounding technic, using GNSS L band signals. In an occultation event, the GNSS signal penetrates the earth's atmosphere and then received by a low earth orbit (LEO) satellite. The signal bends during penetration due to the atmospheric refractivity gradients—a function of atmospheric parameters such as pressure, temperature and water vapor, and thus the bending angle of the signal relates to the
atmospheric parameters. If the signal's bending angle, the location and the velocity of the LEO satellite and GNSS satellite are known simultaneously, the atmospheric parameters can be retrieved. RO technic has many advantages, such as the global coverage, the high vertical resolution and the long-term stability, making up for the deficiencies of traditional observing methods. Besides, the core region of RO technic for atmosphere sounding is 7-25 km which perfectly matches the range of tropopause height. Thus, the advent of global navigation satellite system (GNSS) radio occultation (RO) technic brings an
efficient way for monitoring tropopause. Until now, several occultation missions have been implemented, such as Denmark's Ørsted (Neubert et. al. 2001), Germany's Challenging Mini-Satellite Payload (CHAMP) (Reigber et. al. 2002) and TerraSAR-X (Werninghaus and Buckreuss 2009), Argentina's Scientific Application Satellite C satellite (SAC-C) (Colomb and Varotto 2003), European Meteorological Satellite Organization's Meteorological Operational Satellite Program (MetOp) (Buemi and Caujolle 2008; Dieter Klaes et. al. 2013), US and Taiwan's joint work Constellation Observing System for Meteorology,
Ionosphere & Climate (COSMIC) (Anthes et. al. 2008), China's FunYun3 (Sun et. al. 2018) and so on, providing substantial and valuable data for climatologic and meteorology research.

To detect the global tropopause based on RO technic, there are two mainstream approaches, the lapse rate method according to WMO 1957 and the bending angle method raised by Lewis 2009. The former method is widely used in many tropopause researches (Foelsche et. al. 2008; Schmidt et. al. 2008; Rieckh et. al. 2014; Li et. al. 2017; Liu et. al. 2019), because it is simple
and easy to implement. The latter method has the advantage for it retrieves the tropopause directly from climate benchmark observation, and thus this method shows attraction to relative researchers (Zhang et. al. 2014; Schmidt et. al. 2010). the results





depend on the algorithm parameter setting and few of researches have analysed the difference between the results of bending angle tropopause and lapse rate tropopause in detail.

In this paper, we give the results and bias analysis of tropopause height derived from RO data based on lapse rate method and
bending angle method, respectively. Bending angle profile is level-1 data but dry temperature profile is level-2 data, and thus we only discuss the tropopause height (TPH). Besides, European Centre for Medium-Range Weather Forecasts (ECMWF) do not provide bending angle profiles, and thus we take ECMWF lapse rate tropopause (LRT) height (LRTH) as reference. The structure of this paper is as follow: Sect. 2 gives the introduction about the two tropopause determination methods and the data we used in this research. Sect. 3 presents the results of RO LRTH and the bias between RO LRTH and its collocated ECMWF
LRTH. Similarly, Sect. 4 shows the results of RO bending angle tropopause (BAT) height (BATH) and the bias between RO BATH and ECMWF LRTH. Sect. 5 compares two tropopause definition methods via the comparison of RO BATH and corresponding RO LRTH. Sect. 6 provides a summary.

## 2 Method and Data

### 2.1 Lapse rate method

Lapse rate method is an efficient way to calculate thermal tropopause, which is based on the tropopause definition introduced by WMO 1957. According to the WMO 1957, the tropopause height is defined as the lowest level at which the lapse rate decreases to 2℃/km or less, provided also the average lapse rate between this level and all higher levels within 2km does not exceed 2℃/km. In our algorithm, we first interpolate dry temperature profiles, including both RO and ECMWF data, with 100 m interval to avoid the bias caused by different vertical resolution in some degree. The parameter setting refers to previous
researches of tropopause. The bottom limit and the top limit are set as follow:

$$TPH_{min} = 2.5 \times (3 + \cos(lat \times 2))km \tag{1}$$

$$TPH_{max} = 2.5 \times (7 + \cos(lat \times 2))km \tag{2}$$

$TPH_{min}$ and $TPH_{max}$ are the bottom limit and top limit for the TPH program, respectively. $lat$ is the latitude of the atmosphere profile. Figure 1 gives the sketch map of lapse rate method, in which the blue line presents the temperature gradient, and the
lowest point that temperature gradient larger than -2K/km is determined as TPH point.



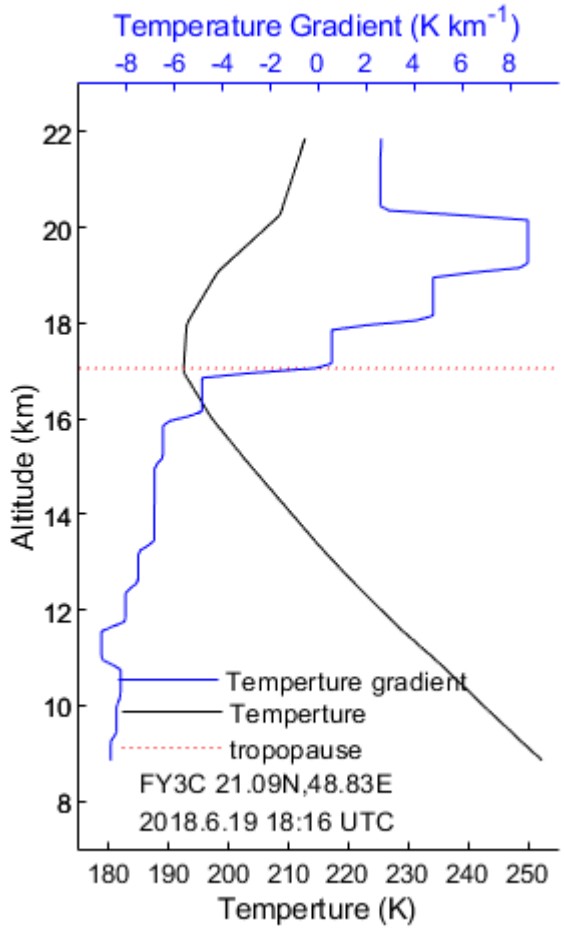

**Figure 1 Sketch map of lapse rate method.**

## 2.2 Bending angle method

Bending angle method introduced by Lewis 2009 uses covariance transform to get the TPH, in which the TPH is defined as

the maximum of covariance transform (CT) of the logarithm of the bending angle. The mathematical expression of covariance

transform is as follow:

$$CT(b) = \frac{1}{2a} \int_{z_b}^{z_t} f(z) h(\frac{z-b}{a}) dz \qquad (3)$$

$$h\left(\frac{z-b}{a}\right) = \begin{cases} f(z) - f(b) \, , \max\left(z_b, b-a\right) < z < \min\left(z_t, b+a\right) \\ \qquad 0 \, , \text{ elsewhere} \end{cases} \qquad (4)$$

$z_t$ and $z_b$ are the upper and lower limit of the data profile. $f(z)$ is the natural logarithm of bending angle at impact parameter

z, and 2a is the width of the covariance transform.

The lower and upper limit of TPH, $TPH_{min}$ and $TPH_{max}$, are set as that in lapse rate method and the 2a is fixed at 25 km, and

Fig. 2 gives an example of BA method.



Based on the combination of value and trend of the logarithm of bending angle and the function 3 and 4, it is easily to see that CT should monotonic increase with the decline of Impact parameter when b is larger than $z_t - a$ because the interval [z, min $(z_t, b + a)$] contributes positive value to CT while the interval [max $(z_b, b - a)$,z] provides negative contribution to CT , and thus, with the extension of the positive interval, the CT will increase steadily. It can be seen in Fig. 2, the CT decreases with the decrease of impact parameter beyond $z_t - a$. This means that $z_t - a$ must higher than the highest TPH to avoid to misjudge TPH point. We set the $z_t$ as 35 km to reduce the amount of computation.

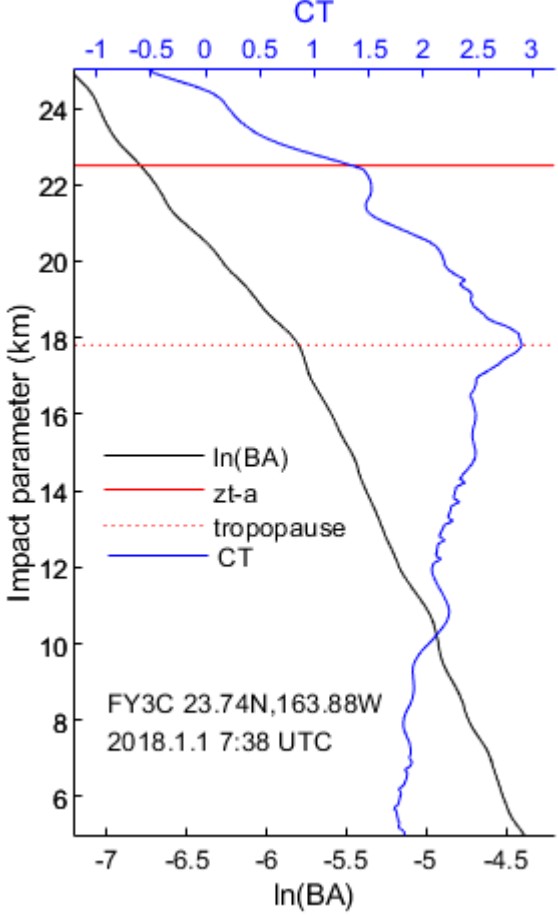

**Figure 2. Sketch map of Bending angle method.**

If the maximum point of CT fulfils the request that the value of this point is at least 5% larger than the average CT over 5 km above and below it (If this point reaches the boundary, it needs to meet one condition). We calculate the satisfied maximum point and the sub-maximum point of each profile for the following discussion. Similar to lapse rate method algorithm, we interpolate the logarithm of bending angle with 100 m interval to ease the effect of different vertical resolution.



## 2.3 Data

To discuss tropopause height derived from RO data and its bias analysis, we use dry temperature profiles and bending angle profiles of two missions, FY3 and MetOp, during 2017.12-2018.2 (DJF) and 2018.6-2018.8 (JJA) for analysing the seasonal difference. We also use ECMWF reanalysis data during corresponding time span to retrieve tropopause height via lapse rate method for reference. The introduction of each data resource is as follow.

### 2.3.1 FY3C RO data

FengYun3 (FY3) Global Navigation Occultation Sounder (GNOS) is operated by Nation Space Science Centre, Chinese Academy of Sciences (NSSC, CAS). GNOS is the first radio occultation sounder that compatible for both GPS and BDS signals for sounding atmosphere and ionosphere (Wang et. al 2014; Sun et. al. 2018). Currently, two satellites, FY3C and FY3D, equipped with GNOS are in orbit, receiving about 500 GPS occultation events and 230 BDS occultation events each day, respectively. Atmospheric profiles retrieved from FY3 GNOS RO data have been accredited by ECMWF, European Organization for the Exploitation of Meteorological Satellites (EUMETSAT) and Wegner Centre for Climate and Global Change (WEGC), and assimilated into numeric weather prediction (NWP) model since 2016. FY3 GNOS ionospheric profiles are also widely used in ionosphere-relative research field.

In this paper, we use dry temperature profiles and bending angle profiles of FY3C GNOS RO data. The amount of FY3C atmospheric profiles is about 400-500 per day, totally 28702 during 2017.12 to 2018.2 and 29814 during 2018.6 to 2018.8, and the vertical resolution of FY3C atmospheric profiles is of 70-100 m.

### 2.3.2 MetOp RO data

MetOp is a series of three polar orbiting meteorological satellites developed by the European Space Agency (ESA) and operated by EUMETSAT. Until now, the three MetOp satellites, MetOp-A (Buemi and Caujolle 2008), MetOp-B (Dieter Klaes et. al. 2013) and MetOp-C, are all in orbit, majoring in providing atmospheric profiles for numeric weather prediction. MetOp has tropopause products, providing latitudinal mean tropopause height each month, which can be found on www.romsaf.org.

In this work, dry temperature profiles and bending angle profiles retrieved from MetOp-A and MetOp-B RO data are download from www.romsaf.org. Due to the number of RO sounding satellites, the amount of used MetOp atmospheric profiles is about 1200 per day, totally 69030 in the span of 2017.12 to 2018.2 and 76992 for 2018.6 to 2018.8. The vertical resolution of MetOp atmospheric profile is similar with FY3C counterpart, about 100 m.



### 2.3.3 ECMWF reanalysis data

With respect to ECMWF reanalysis data, these data downloaded from ECMWF operational archive, recording the global

temperature at 0:00, 6:00, 12:00 and 18:00 per day with 137 model levels and 128×64 longitude×latitude grids are used in our study. The vertical resolution of ECMWF reanalysis data is coarse than two RO missions, decreasing from 200 m at 0 km to 300 m at 20 km.

### 3 Results and bias analysis of RO LRTH

In this section, we take ECMWF LRTH as reference (true value) to give the bias regulation of RO LRTH during DJF and JJA,

for ECMWF data provide no bending angle profile.

As mentioned above, ECMWF reanalysis data records the global atmosphere profile 4 times a day, with the time interval of 6 hour, and the global map is evenly divided into 128×64 longitude×latitude grid, which means the distance between adjacent grid points is about 200-300 km in terms of latitude. Based on the above-mentioned condition, we choose the ECMWF atmosphere profile with the smallest temporal and spatial interval with each RO profile as the collocated ECMWF data.

Figure 3 depicts the data distribution and the RO LRTH result and the bias between RO LRTH and their collocated ECMWF LRTH during 2017.12-2018.2. The data distribution of FY3C and MetOp are almost the same, for they are both polar orbiting satellites. Holistically, the top two panels of Fig.3 indicate that LRTH results derived from FY3C data and MetOp data conform to each other. Considering the bias between RO LRTH and ECMWF LRTH, both FY3C LRTH and MetOp LRTH have the same bias over 40N-90N, about average ±0.1 km. Over mid latitude range, the bias of FY3C and MetOp are different. Many

FY3C tropopause points show positive bias while the MetOp counterparts show negative bias. Over Antarctica, it seems that most FY3C LRTH has negative bias in the range of 0.2 km to 0.4 km compared with ECMWF results, but MetOp LRTH presents similar results with ECMWF LRTH.

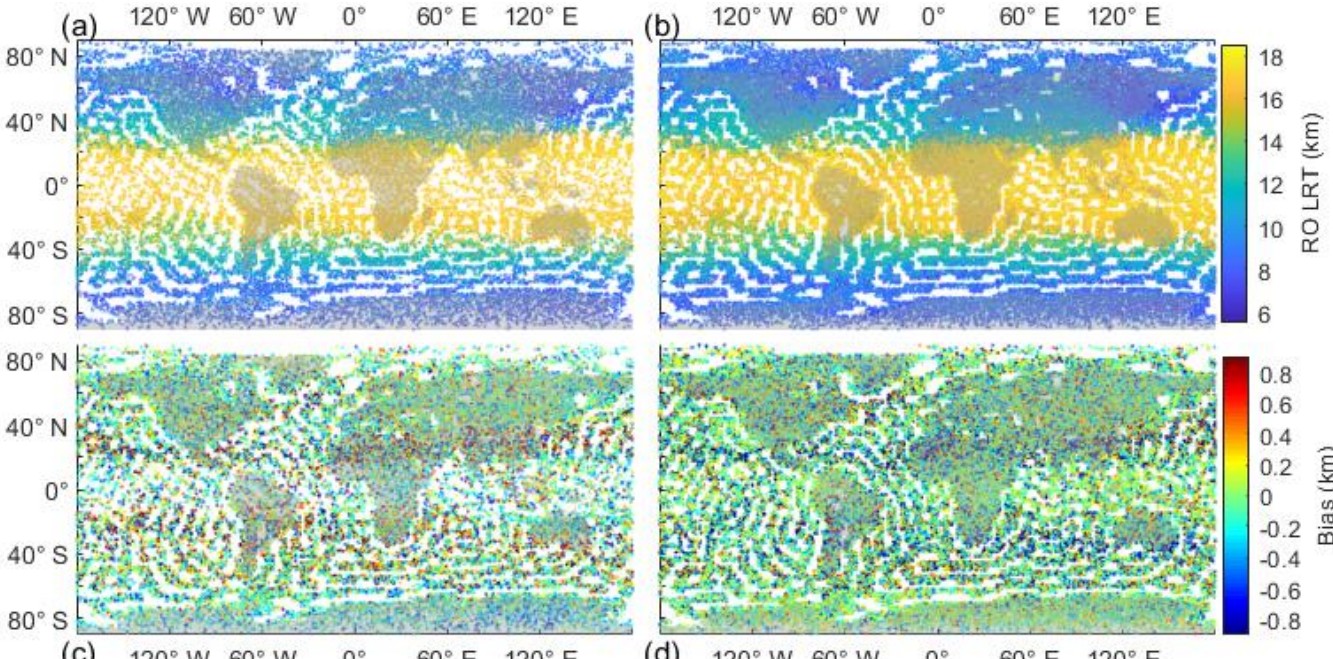

**Figure 3. Global LRT derived from two RO mission data and bias between RO LRTH and ECMWF LRTH (RO-ECMWF) during**
**2017.12-2018.2. (a), (b), (c), (d) for FY3C LRT results, MetOp LRT results, bias between FY3C results and ECMWF results, bias**
**between MetOp results and ECMWF results, respectively.**

To detailly analyse the latitudinal bias distribution, we divide the 90S-90N into 18 latitude bands centred at 85S to 85N. The
LRTH bias distribution of each latitude band, including the mean bias and root mean square error (RMSE) between FY3C
LRTH and ECMWF LRTH, derived from FY3C and MetOp data in the span of 2017.12 to 2018.2 is illustrated in Fig. 4. Two
similar RMSE curves indicate that the magnitude of bias between two RO LRTH and their collocated ECMWF LRTH is
similar. The bias mainly occurs over mid latitude regions and is more prominent over 20° N-30° N. In north hemisphere, the
RMSE surges over 5° N (0.5 km) to 25° N (1.5 km), and then decreases sharply to 0.5 km at 45° N. In contrast, variation of
RMSE in south hemisphere is much milder. Although FY3C and MetOp share a common RMSE distribution, their mean bias
against their collocated ECMWF LRTH present an opposite trend. MetOp LRTH exhibits apparent negative bias over the
vicinity of 25° N and 35° S but 25° N and 35° S are the maximum points of mean bias of FY3C LRTH. Besides, over Antarctica,
LRTH from FY3C shows an obvious negative bias while MetOp LRTH matches well with ECMWF result.



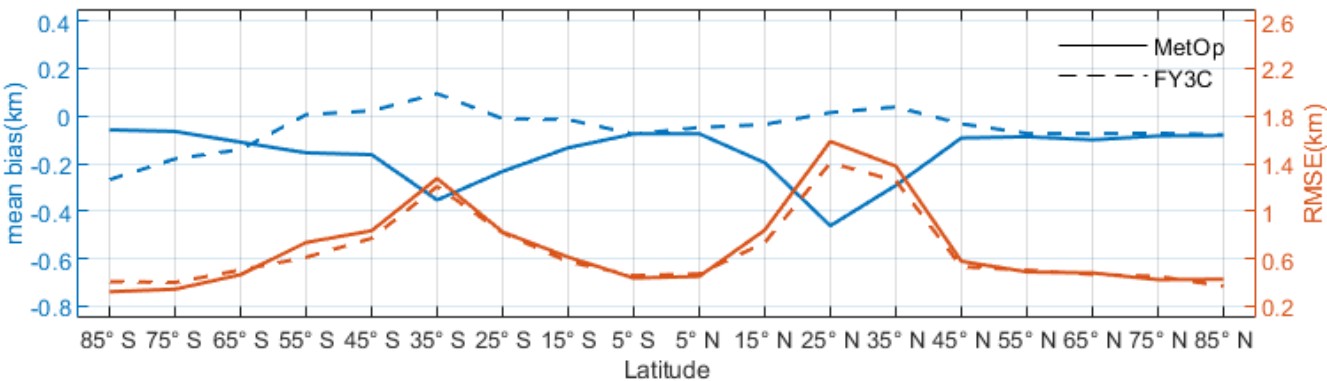

**Figure 4 Latitudinal bias distribution between RO LRTH and ECMWF LRTH (RO-ECMWF) during 2017.12-2018.2**

Similar to Fig. 3-4, Fig. 5-6 introduce the global RO LRTH results, global bias distribution and the latitudinal bias

distribution between RO LRTH and their corresponding ECMWF LRTH during JJA. Compared with DJF results, the LRTH

results and the bias regulation during JJA is totally different. Similar with Fig.3, Fig. 5 illustrates the RO LRT results and the

bias between RO LRTH and ECMWF LRTH during JJA in 2018. For the LRT results, both two RO LRT results show that

the high tropopause region (yellow region in top two panels) broaden northward, and tropopause height over Antarctica

raises obviously, compared with the DJF results. For the bias between RO LRTH and ECMWF LRTH, there are still

apparent bias over mid latitude for both two RO results, which is similar with the DJF results, but over Antarctica, both

FY3C and MetOp LRT show distinct negative bias, compared with ECMWF result, which is different from DJF results.

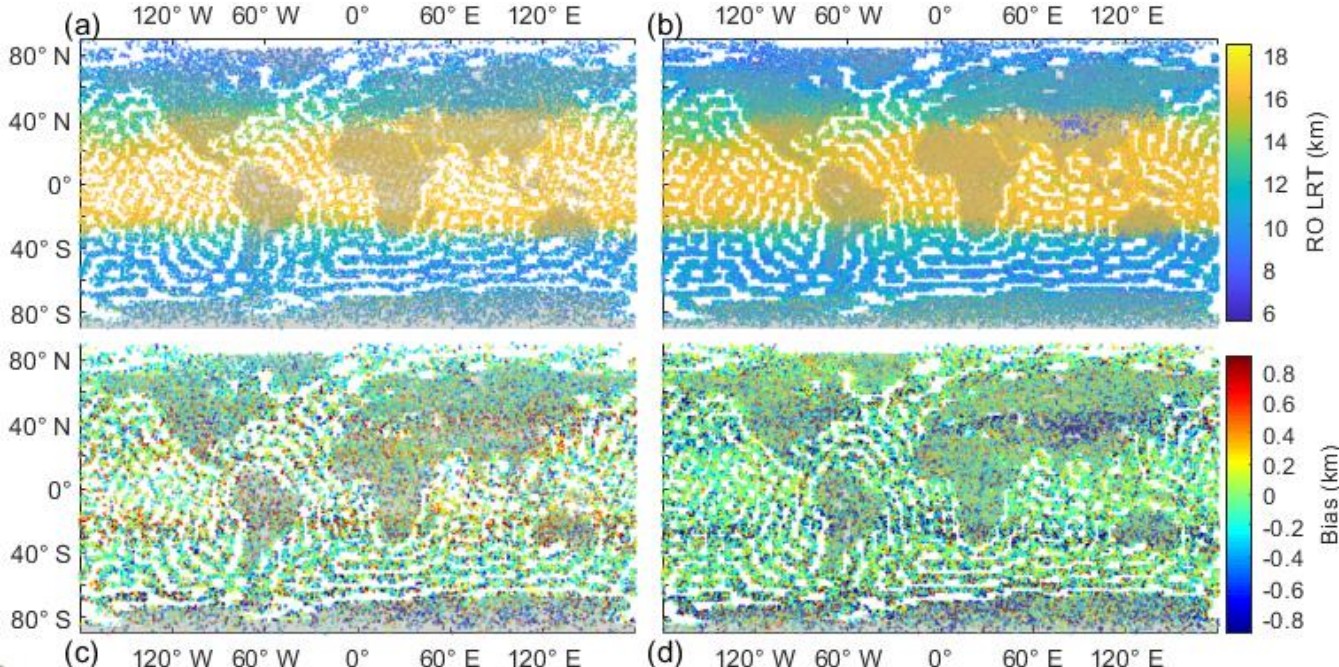





**Figure 5 Global LRT derived from two RO mission data and bias between RO LRTH and ECMWF LRTH (RO-ECMWF) during 2018.6-2018.8. (a), (b), (c), (d) for FY3C LRT results, MetOp LRT results, bias between FY3C results and ECMWF results, bias**
**between MetOp results and ECMWF results, respectively.**

Figure 6 demonstrates the latitudinal bias distribution of FY3C and MetOp LRTH during 2018.6-2018.8, which is significantly different with DJF LRTH latitudinal bias distribution. For RMSE distribution, except for the two peaks over mid latitude regions in south and north hemisphere, a new peak over Antarctica appears during JJA. The interval of two mid latitude peaks widens obviously, from 10° latitude (5° S-5° N) to more than 20° latitude (at least 5° S-15° N). Compared
with DJF result, in north hemisphere, FY3C RMSE decreases prominently during JJA, but the value of MetOp RMSE is almost unchanged. The MetOp mean bias curve in Fig. 6 is similar with that in Fig. 4, except for the negative bias over Antarctica which is not shown in DJF. However, the negative bias of FY3C LRTH during JJA is smaller than that in DJF, and the mean bias curve has only one peak over 25S and decreases linearly from 25S to the north pole.

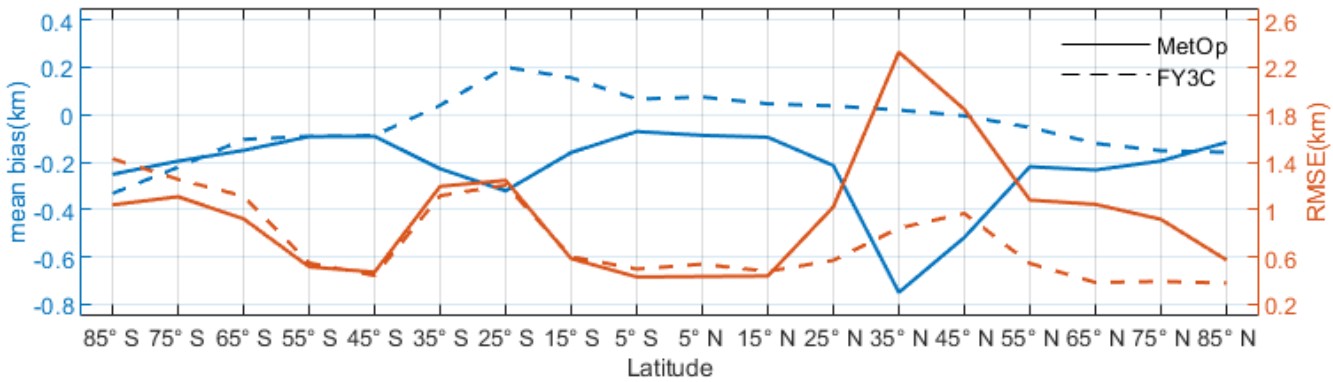

**Figure 6 Latitudinal bias distribution between RO LRTH and ECMWF LRTH (RO-ECMWF) during 2018.6-2018.8**

**4 Results and bias analysis of RO BATH**

In this section we discuss the regulation of bias between RO BATH and their collocated ECMWF LRTH. Like section 3, we first give the BAT results derived from FY3C and MetOp data, and their bias compared with collocated ECMWF LRT. Figure 7 and 8 depict the BAT results and BATH bias during DJF and JJA, respectively. On the whole, compared with ECWMF
LRTH, positive bias, about 0.6 km to 1 km is shown globally on RO BATH results during both two seasons. Like the bias between RO LRTH and ECMWF LRTH in section 3, obvious bias occurs over two mid latitude band during DJF and an extra Antarctica region during JJA. However, unlike the LRTH bias regulation that RO LRTH shows relatively consistent positive or negative bias over those specific regions, for RO BATH, positive bias and negative bias are mixed over those regions, especially for the span of June to August.



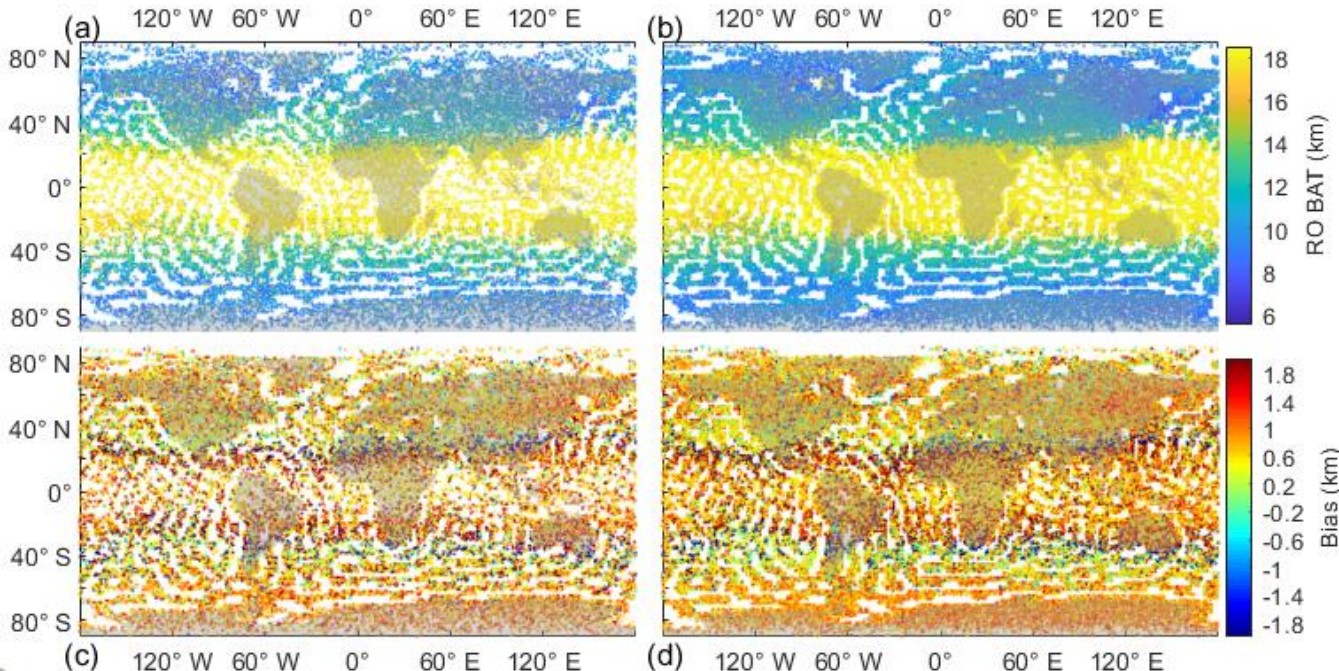

**Figure 7 Global BAT derived from two RO mission data and bias between RO BATH and ECMWF LRTH (RO-ECMWF) during 2017.12-2018.2. (a), (b), (c), (d) for FY3C BAT results, MetOp BAT results, bias between FY3C results and ECMWF results, bias between MetOp results and ECMWF results, respectively.**

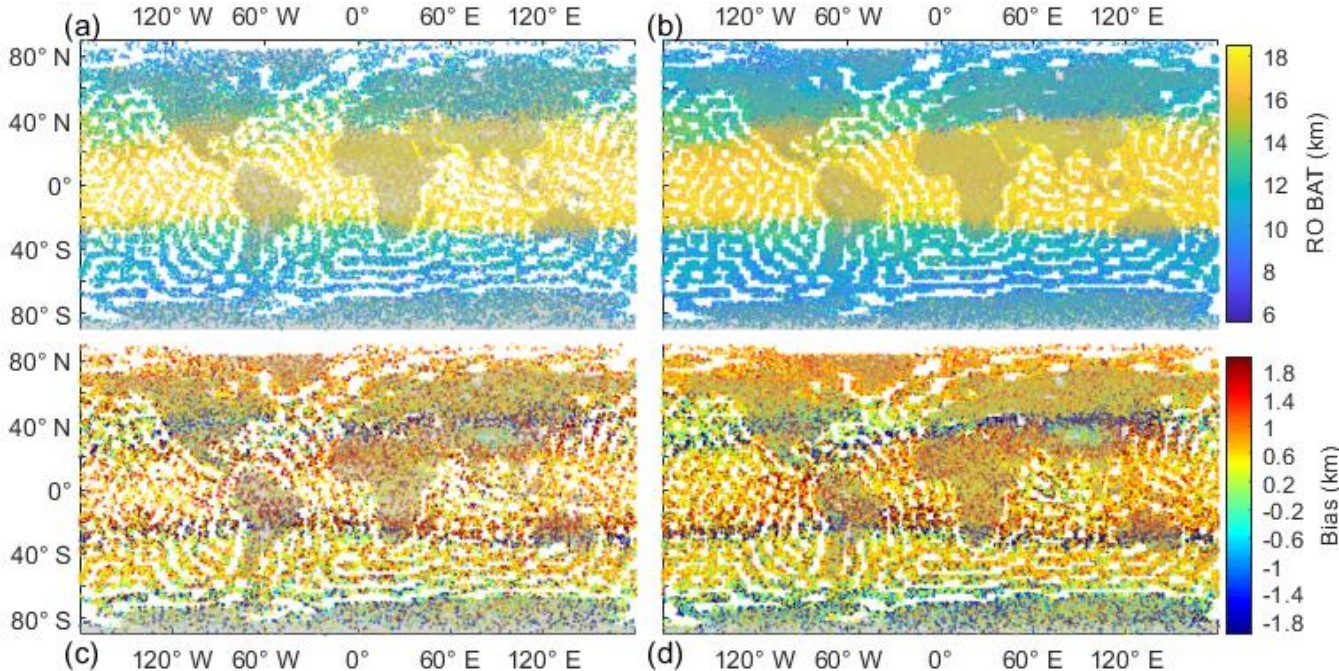





**Figure 8 Global BAT derived from two RO mission data and bias between RO BATH and ECMWF LRTH (RO-ECMWF) during 2018.6-2018.8. (a), (b), (c), (d) for FY3C BAT results, MetOp BAT results, bias between FY3C results and ECMWF results, bias between MetOp results and ECMWF results, respectively.**

Also, we give the latitudinal bias regulation. Fig. 9 and 10 show the latitudinal bias distribution of BATH derived from FY3C GNOS and MetOp data during DJF and JJA, respectively. Both FY3C GNOS BATH and MetOp BATH have the similar mean

bias trends, and they both show different symmetrical characteristics during two seasons. The means bias distribution is of axial symmetry at 0° during DJF but of approximately point symmetry at 20° S during JJA. For the RMSE, the RMSE trend of MetOp BATH is similar with that of MetOp LRTH presented in Fig. 4 and Fig. 6. RMSE of FY3C GNOS BATH is prominent higher than RMSE of MetOp BATH over high latitude regions during DJF, especially for Antarctica where the RMSE of FY3C is 0.6 km higher than that of MetOp. The large bias disappears during JJA, but RMSE of FY3C results appears

increasing RMSE over 45N to 90N compared with MetOp results.

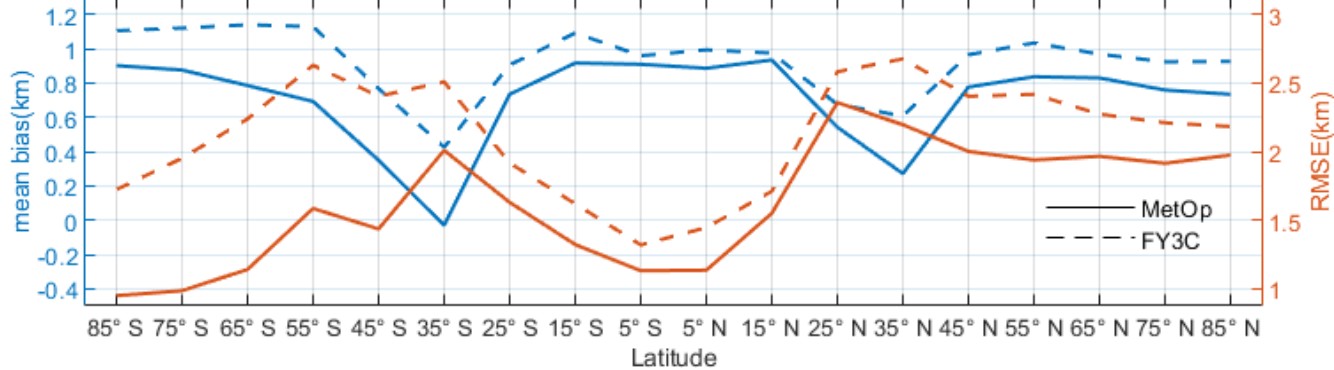

**Figure 9 Latitudinal bias distribution between RO BATH and ECMWF LRTH (RO-ECMWF) during 2017.12-2018.2**

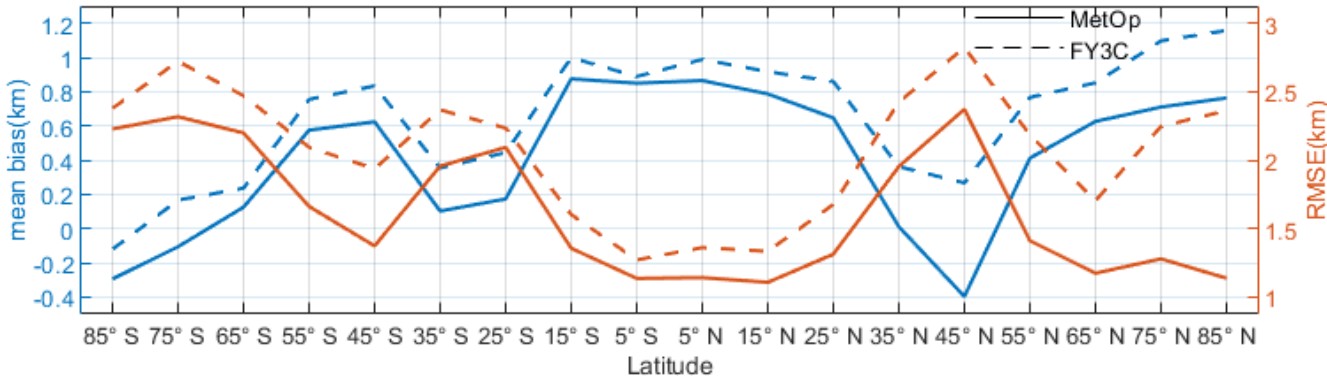

**Figure 10 Latitudinal bias distribution between RO BATH and ECMWF LRTH (RO-ECMWF) during 2018.6-2018.8**



## 5 Comparison between RO BATH and RO LRTH

In above sect. 3 and sect. 4, we take ECMWF LRTH as the true value for tropopause height to analyse the absolute bias of RO LRTH and RO BATH. As a supplement, comparison results of different methods based on corresponding RO data are provided in this section for objective assessment of the difference between LRTH and BATH.

The top panel of Fig. 11 and 12 show the tropopause height calculated by above two methods during DJF and JJA based on FY3C and MetOp data, respectively, while the mid and bottom panels of two figures illustrate the latitudinal mean bias and RMSE between BATH and LRTH. The results derived from two RO mission's data show overall consistency. It is obvious that BATH is globally higher than LRTH except for mid latitude region during two seasons and Antarctica during JJA. Over tropics, the mean bias between BATH and LRTH is almost constant during two seasons, and RMSE is also small, indicating that BATH is consistently higher than LRTH. The mean value of BATH and LRTH meet each other at 10° latitude outside the tropics, but the corresponding RMSE is high, which means that positive bias and negative bias counteract the effort of each other. Over Antarctica, during DJF, the mean bias is large but the RMSE is relatively small, but during JJA, the situation is opposite, and notably, the RMSE of MetOp results during DJF is extremely small. However, although there is obvious bias between BAT and LRT, their variation trends are almost the same. Both BATH and LRTH indicate that the gradient of TPH from 25° S to 65° S is milder than that from 25° N to 65° N during DJF, but sharper during JJA, and the cap of TPH during JJA is wider and slightly lower than that during DJF. On the whole, the global TPH derived from both methods has a 10°





southward shift from DJF to JJA, but is constant over Arctic and TPH over Antarctica higher during JJA.

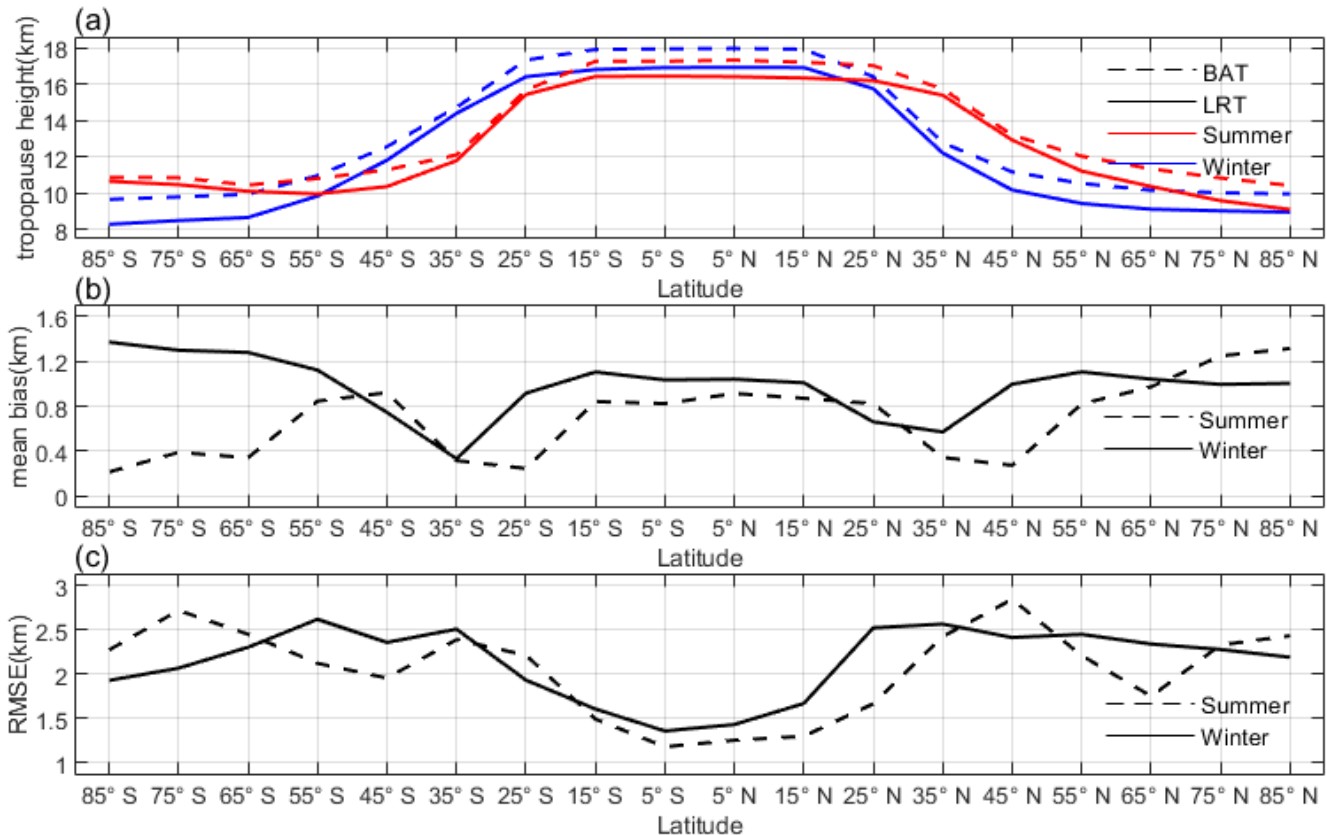

**Figure 11 Comparison results between BATH and LRTH derived from FY3C data during DJF and JJA. (a), (b), (c) for Latitudinal tropopause results, mean bias between BATH and LRTH (BATH subtract LRTH), and RMSE between BATH and LRTH, respectively.**






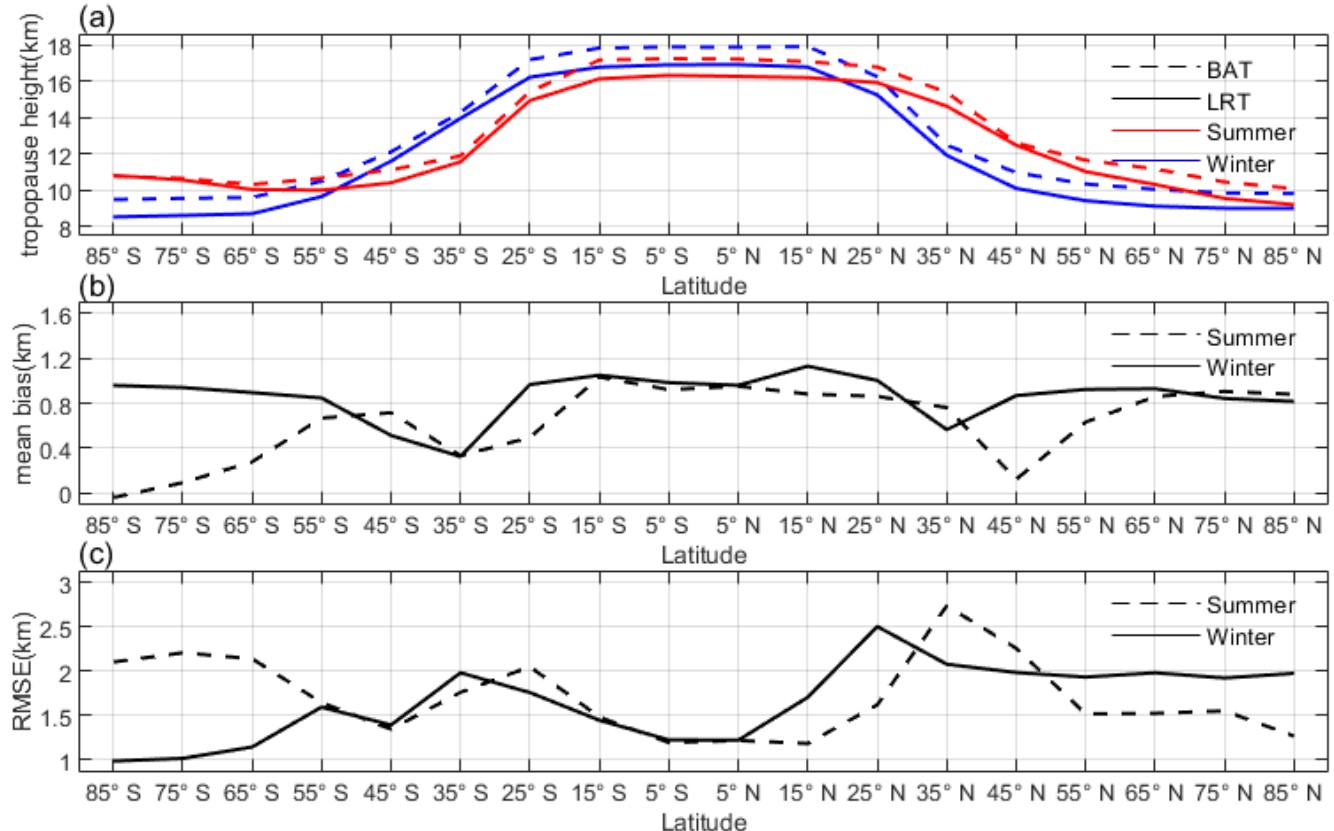

**Figure 12 Comparison results between BATH and LRTH derived from MetOp data during DJF and JJA. (a), (b), (c) for Latitudinal tropopause results, mean bias between BATH and LRTH (BATH subtract LRTH), and RMSE between BATH and LRTH, respectively.**

**6 Summary**

Tropopause is a significant region among the earth's atmosphere, whose variation trend is close related to the climate change, now becoming a hot topic in atmosphere research field. To detect the global tropopause, there are mainly two robust methods, the temperature lapse method introduced by WMO 1957 and the bending angle covariance transform method raised by Lewis 2009. This paper takes LRTH derived from ECMWF reanalysis data as reference to analyse the bias regulation of both RO

LRT and RO BAT and gives the comparison between RO BATH and RO LRTH.

For the bias regulation of RO LRTH, compared with ECMWF LRTH, the most obvious bias occurs at 35° S and 25° N during DJF but 25° S and 45° N during JJA. However, FY3C LRTH show positive bias compared with ECMWF results while MetOp LRTH show negative bias. With respect to BATH, compared with LRTH, obvious positive mean biases, about 0.8-1.2 km, are shown over tropics and high latitude region (only DJF for Antarctica), while over mid latitude (two seasons) and Antarctica



(JJA), the difference between each BATH point is quite large, but counteracts to each other, showing less mean bias. The curve of mean bias between BATH and LRTH shows different symmetric features for different seasons. However, both BATH and LRTH exhibit the similar variation trend, emphasizing the availability of both two methods.

**Data availability**

FY3C RO data are available from National Satellite Meteorological Center ([www.nsmc.org.cn](www.nsmc.org.cn)) and MetOp RO data are
available from the Radio Occultation Meteorology Satellite Application Facility ([www.romsaf.org](www.romsaf.org)). The ECMWF reanalysis data are available from ECMWF website ([apps.ecmwf.int/archive-catalogue/?class=od&stream=oper&expver=1&type=an](apps.ecmwf.int/archive-catalogue/?class=od&stream=oper&expver=1&type=an)).

**Author contributions**

Conceptualization, Z.L. and W.B.; Data curation, J.X., D.Z. and C.L.; Funding acquisition, Y.S., C.C., Q.D., X.W., X.M., Y.C. and D.W.; Resources, Y.S.; Calculation program, Z.L., G.T. and Y.T.; Writing – original draft, Z.L.;
Writing – review & editing, W.B..

**Competing interests.**

The authors declare that they have no conflict of interest.

**Acknowledgements**

We thank ROM SAF group for providing free-accessed MetOp data, and we also show appreciation to ECMWF organization for their grided reanalysis data.

**Financial support**

This research was supported by the National Natural Science Foundation of China (Grant Nos. 41405039, 41775034, 41405040,
41505016, 41505030 and 41606206), the Strategic Priority Research Program of Chinese Academy of Sciences (Grant No. XDA15007501), the Scientific Research Project of the Chinese Academy of Sciences (Grant No. YZ201129), and the Feng Yun 3 (FY-3) Global Navigation Satellite System Occultation Sounder (GNOS) development and manufacture project led by NSSC, CAS.

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
