# Peer review of "Comparison of RO tropopause height based on different tropopause determination methods"

_Atmospheric Measurement Techniques, 2019_

## Referee Comment (RC1) · Anonymous Referee #1 · 13 Dec 2019

**Context and general comments**

The paper describes a comparison between two methods for retrieval of tropopause height from GPS radio occultation (RO) data: a) a method using the WMO tropopause definition based on the temperature lapse rate, and b) a method based on the co-variance transform of the logarithmic bending angle profile. The latter method was designed [Lewis, 2009] to use RO data at an earlier stage in the RO processing chain, thus avoiding potential biases introduced by a priori information and an assumption about hydrostatic equilibrium. Collocated tropopause heights from the ECMWF operational archive, based on temperature lapse rates, are used as a reference for comparison.

Comparisons are done between RO and ECMWF data, separately for the two methods

and for data from Metop and from FY3C. The geographical distributions of biases are plotted. In addition, the differences between the two methods are studied, separately for Metop and for FY3C.

Comparisons between the two methods have been described before, particularly in the publication [Lewis, 2009] where the bending angle method was first described. However, the present study gives a more detailed description of the statistical distribution of the differences in latitude and across geographical regions. In addition, it provides some new information on the differences between observations made from Metop and from FY3C. A part of those differences could be caused by the different local times of the two LEO satellites, but particularly for the method based on temperature lapse rates it is not unlikely that different processing play a role (see below).

The manuscript is easy to follow and some new interesting observations on the differences between the methods and the differences between Metop and FY3C are presented. The analysis is fundamentally sound but leaves some important gaps, of which the most important is that the paper lacks a discussion of the findings and that some important facts are not clearly pointed out as they ought to be. In particular, the latitudinal bias structures (i.e., the biases between RO and reanalysis) are relatively similar for the two methods in Metop data, but distinctly different in FY3C data. Figs. 4 and 6 shows that the bias structures based on the temperature lapse rates are very different for the two satellite missions. This is most likely an indication that the processing from bending angle to temperature data, which according to Section 2.3 has been obtained from two different archives, have differences that affect the conclusions. Or could it be the fact that Metop and FY3C sample the atmosphere at different local times? The observation that the bias structures are relatively similar in bending angle provides an argument against this. An important question is thus what we can infer from the differences between the two methods, when these differences depend on which data set we use. A reader of the paper wouldn't know from the facts presented in the paper, it is not discussed, and the problem is not adequately pointed out. This is an important

weakness of the paper.

A publication of this paper would require that the above mentioned issue is analysed and discussed. What are the potential cause of the differences between Metop and FY3C and how does it affect the conclusions?

The English language in the manuscript is not of sufficient quality (note to Editor: does it comply with the standards of this journal?). It needs to be improved before publication. Some of the figures, in particular the latitude-longitude plots have a too low resolution.

**Specific questions to the authors**

1. Writing proper English for a non-native speaker can be difficult. I can only recommend that the authors ask someone fluent in English to have a closer look at the text, and suggest updates to the text.

2. The figures showing the coloured latitude-longitude plots need to have a higher resolution.

3. Are the ECMWF data taken from the ERA-Interim reanalysis? If so, it should be stated. Or is it operational NWP analysis data that are used?

4. Can the authors think of reasons for the bias structures in LRTH to be so different for Metop and FY3C (even if the biases are relatively small in absolute numbers, they are structurally significant)? Are there systematic differences in the set of temperature profiles? After all, the bias structures in BATH are much more similar between Metop and FY3C.

---

## Author Comment (AC1) · 2 Jan 2020

Dear reviewer: Thank you very much for your comments and advices on our manuscript. We have carefully read your comments and revised our manuscript. The reply with figures in PDF version is uploaded in zip file, and our response to your comments is as follows: Context and general comments 1. "A part of those differences could be caused by the different local times of the two LEO satellites, but particularly for the method based on temperature lapse rates it is not unlikely that different processing play a role" Our response: MetOp-a, MetOp-b and FY3C satellite are all sun-synchronous orbit satellites, which makes them have the similar local solar time, like fig. 1, 2 and 3. These figures are generated via ROM SAF website: https://www.romsaf.org/monitoring/index.php Here we only present one-month local

solar time distribution for example. Also, MetOp and FY3C RO data are both processed via ROPP software (from excess phase to bending angle, refractivity, dry temperature, etc). Thus, we use FY3C and MetOp data in our study to avoid inter-mission bias as far as possible. We add a description about this in section 2.3 in our updated manuscript. However, the bias is inevitable because the RO sounders are different and the version or parameters of ROPP software may be different. The retrieval from raw RO data to bending angle and temperature is complex, so, it's difficult for us to explain why FY3C LRT TPH and MetOp LRT TPH are different over mid latitude, clearly. However, we show the latitudinal bias characteristics of the results from both two determination methods and two RO missions' data, respectively, which is a meaningful reference for relative researchers using these two RO mission's data. 2. "In particular, the latitudinal bias structures (i.e., the biases between RO and reanalysis) are relatively similar for the two methods in Metop data, but distinctly different in FY3C data." Our response: To consider the results of different methods for FY3C and Metop individually, we would like you to see the fig.11(two methods comparison for FY3C) and fig.12 (Metop), where we can see that both results and bias structures are similar. (ie. The mean bias curve has significant seasonal difference over Antarctic, and the minimum point over north hemisphere has a 10-degree northern shift from winter to summer.) In addition, we think the bias structures of LRT TPH (bias between RO LRT TPH and ECMWF LRT TPH) and the bias structures of BA (Bending Angle) TPH (bias between RO BA TPH and ECMWF LRT TPH) should be studied separately, because the bias scale is different and principles of two methods are also different. Moreover, because we don't have authoritative bending angle profiles (like ECMWF temperature profiles), and thus the bias structures can only present as bias between RO BA TPH and ECMWF LRT TPH instead of bias between RO BA TPH and authoritative BA TPH. 3. "Figs. 4 and 6 shows that the bias structures based on the temperature lapse rates are very different for the two satellite missions. " Our response: As we have explained in response 1. We have excluded the effect of local solar time and fig.3,5,7,8 in our paper indicate the same data global distribution of two RO missions. For the quality of RO data, the ROM

SAF website (We have mentioned above) gives the global BA and refractivity statistics for both MetOp and FY3C mission (following fig.4-6, we only present BA here), showing remarkably consistent mean and std curves and this indicates that the both data products are qualified. Above all, we exclude the reason of local solar time, data global distribution and the quality of RO data. However, the inner process of retrieval from RO signal to temperature is complex, and thus we cannot answer the reason of the mid-latitude difference clearly now. This need further study. 4. "An important question is thus what we can infer from the differences between the two methods, when these differences depend on which data set we use." Our response: Like response 1 and 2, we would like you focus on fig.11 and 12 to consider the differences between the two methods as the inter-mission bias is inevitable. Fig.11 and 12 indicate the different TPH results of two methods of FY3C data and MetOp data, respectively. The bias structures and the results from two RO missions are similar. For the purpose of our paper, we would like to gives comparison results for researchers of this field. They will choose the proper methods fitting for their study. Tropopause do not have a specific definition, and thus we cannot say which method is better. Although the results are different, results from both methods reflect similar trend, and we would like to show the trend instead of evaluating which trend is correct. Specific questions to the authors 1. "Writing proper English for a non-native speaker can be difficult. I can only recommend that the authors ask someone fluent in English to have a closer look at the text, and suggest updates to the text." Our response: We have asked for edit service to revise our manuscript. 2. "The figures showing the coloured latitude-longitude plots need to have a higher resolution." Our response: We are sorry for this. Those figures look good in word file before upload, but are unsatisfactory on web pages. Maybe it is due to changes on the figure's scale. We re-export the figures in tiff format, but we don't know the effect before we upload the figures. If the problem still exists, we will continue modifying. 3. "Are the ECMWF data taken from the ERA-Interim reanalysis? If so, it should be stated. Or is it operational NWP analysis data that are used?" Our response: It is operational NWP analysis data. Those 137 model level data are downloaded from ECMWF operational

archive. We have corrected our word in section 2.3.3. 4. "Can the authors think of reasons for the bias structures in LRTH to be so different for Metop and FY3C (even if the biases are relatively small in absolute numbers, they are structurally significant)? Are there systematic differences in the set of temperature profiles? After all, the bias structures in BATH are much more similar between Metop and FY3C." Our response: We have explained for this in response 1, 2 and 3 for Context and general comments. Why the LRT tropopause from FY3C and MetOp is different over mid latitude is also an interesting question for us. It needs further study.

We hope to hear from you soon. Any advice or comments would be greatly appreciated.

Please also note the supplement to this comment:
https://www.atmos-meas-tech-discuss.net/amt-2019-379/amt-2019-379-AC1-supplement.zip

―――――――――――――――――

[Figure]

[Figure]

**Fig. 1.**

[Figure]

[Figure]

**Fig. 2.**

[Figure]

[Figure]

[Figure]

**Fig. 3.**

BA Global O-B statistics for Metop-A processed by EUMETSAT

QC applied
No. of occultations: 674
Data from 22/08/18 to 23/08/18

Met Office

Number of occultations

— Mean (Met Office)
-- Std dev (Met Office)
-·- Count (Met Office)

EUMETSAT
ROM SAF

ECMWF

Plotted at 06:27, 23 Aug 2018

**Fig. 4.**

[Figure]

[Figure]

**Fig. 5.**

[Figure]

**Fig. 6.**

---

## Referee Comment (RC2) · Anonymous Referee #1 · 10 Jan 2020

**General comment**

I would like to point the authors to the fact that, to my knowledge, the ROPP Fortran package contains forward modelling routines from model pressure, temperature, and humidity data to both refractivity and bending angle. I don't know whether such model bending angles would be suitable for application of the covariance transform, but I suggest that the authors consider that as a possibility. I'm not requiring that you do that for this present study, but it should be considered in future studies.

**Minor remarks**

Abstract, line 11: Replace "The Tropopause is a significant . . ." with "The tropopause

is an important . . ." (lower case 't' in 'tropopause')

Abstract, line 15: From your comments I understood that you do not use reanalysis data.

Abstract, lines 16-19: A suggestion: "We compute biases of the RO lapse rate tropopause height (LRTH) and the RO bending angle tropopause height (BATH) relative to the ECMWF LRTH. The dependences of the tropopause height biases on TPH retrieval method, latitude, season and RO mission are investigated."

Abstract, line 19: Start sentence with "the": "The results indicate . . ."

It should be acknowledged that the main text has undergone a significant improvement since the first version.

On page 3, lines 65-66, I suggest to remove the whole sentence "The bending angle profile is level-1 data . . .". The definition of processing levels is not a scientific issue and is not interesting or important for the reader. Related to this change you should also remove "Moreover" from the beginning of the sentence on line 66.

In several places in the main text you use the phrase "bias regulation". That is not a correct use of the word "regulation". It seems that you just mean "biases"?

On page 7, line 148, I suggest that you remove "(true value)" from the sentence. ECMWF is a reference, and nothing more.

---

## Referee Comment (RC3) · Anonymous Referee #1 · 14 Jan 2020

I have no further comments. The only remaining issue is to make sure that the figures in the final submission have a far better quality than in the present submission.

---

## Author Comment (AC2) · 14 Jan 2020

Dear reviewer: Thanks for hearing from you again. We have revised our manuscript according to your comments and gave the following responses.

General comment:

"I would like to point the authors to the fact that, to my knowledge, the ROPP Fortran package contains forward modelling routines from model pressure, temperature, and humidity data to both refractivity and bending angle. I don't know whether such model bending angles would be suitable for application of the covariance transform, but I suggest that the authors consider that as a possibility. I'm not requiring that you do that for this present study, but it should be considered in future studies."

Our response: We think bending angle computed by ROPP should be suitable for applying the covariance transform, because there is a tropopause package in ROPP Applications module that contains bending angle covariance transform method. Actually, we referred to ROPP tropopause package user guide for setting the parameters in our code (We have uploaded the user guide in supplement file, and you can also find it on https://www.romsaf.org/ropp/). Normally, the ROPP Applications module should be compatible with ROPP forward module. However, the FY3C bending angle and dry temperature profiles are not retrieved by us, and thus we don't know clearly about ROPP. Maybe there are some issues in the link between forward module and application module, and this point is worth to consider. We will do it in further study.

Minor remarks:

Abstract, line 11: Replace "The Tropopause is a significant ..." with "The tropopause is an important ..." (lower case 't' in 'tropopause')

Abstract, lines 16-19: A suggestion: "We compute biases of the RO lapse rate tropopause height (LRTH) and the RO bending angle tropopause height (BATH) relative to the ECMWF LRTH. The dependences of the tropopause height biases on TPH retrieval method, latitude, season and RO mission are investigated."

Abstract, line 19: Start sentence with "the": "The results indicate ..."

On page 3, lines 65-66, I suggest to remove the whole sentence "The bending angle profile is level-1 data ...". The definition of processing levels is not a scientific issue and is not interesting or important for the reader. Related to this change you should also remove "Moreover" from the beginning of the sentence on line 66

On page 7, line 148, I suggest that you remove "(true value)" from the sentence. ECMWF is a reference, and nothing more.

Our response: Thanks for your suggestions. We have corrected our manuscript according to these. The ''true value" is also replaced by ''reference" in line 236.

In several places in the main text you use the phrase "bias regulation". That is not a correct use of the word "regulation". It seems that you just mean "biases"? Our response: In line 149, 206, 264, we replaced the "bias regulation" as "latitudinal biases", and in line 183, we used biases distribution here for it corresponds to global LRT (figure 5 and 6), and in line 262, we use biases for it is a summarization.

There is an attention "please do NOT submit your revised manuscript here as supplement" and thus we don't upload the revised manuscript this time (we don't know the reason why there is a such attention and last time there were large changes in our manuscript and thus we uploaded it). However, if you require the manuscript, we will upload it. We are looking for hearing from you.

Please also note the supplement to this comment:
https://www.atmos-meas-tech-discuss.net/amt-2019-379/amt-2019-379-AC2-supplement.zip

---

## Referee Comment (RC4) · Anonymous Referee #1 · 29 Jan 2020

I have no further comments but to point out the possibility to English language copy-editing support, which to my understanding is offered by the journal.

---

## Author Comment (AC3) · 1 Feb 2020

We thank you again for your comments and advices. For the language editing service, we choose this because it is easy to contact.

---

## Referee Comment (RC5) · Anonymous Referee #2 · 11 Feb 2020

** General comments

The manuscript under review, "Comparison of RO tropopause height based on different tropopause determination methods" by Liu et al., uses a tropopause definition tailored towards the radio occultation (RO) measurement technique–taking advantage of bending angle information which is very close to "pure" RO observations–and compares the resulting tropopause characteristics to the widely used temperature lapse rate definition of the thermal tropopause.

The paper analyzes data from two satellite missions, FY3C and Metop, and compares to collocated ECMWF analysis data using both methods and for two specific seasons. Compared are also the differences between the two missions.

It makes sense to base global tropopause monitoring by RO on such a bending angle-based definition, because for RO the bending angle is the more natural parameter choice compared to temperature, and less contaminated by retrieval details. It is then of great value to see how this definition compares to the common WMO definition. Also, RO is a good choice for global tropopause monitoring due to its properties.

I have two major concerns to be addressed before I would consider the study ready for publication:

* Using RO data from two missions introduces a complication to this study. I see two issues here: First, I think it would help the paper to keep its focus on the comparison of the two tropopause determination methods, instead of also comparing two missions. Secondly, and more importantly, the data of these missions are from two different processing centers and I suppose that they are not consistently processed. Metop data are processed by ROMSAF, and information about the processing of FY3C is missing in the paper. It has been shown in the literature that only for the same processor and processing version profiles from different RO missions can be mixed together. Applying the tropopause determination algorithms on inconsistently processed profiles will most probably result in a systematic bias. I would strongly recommend to only use data from one processing to avoid this additional complication, or to first validate that no bias is introduced.

* Lewis 2009 showed comparisons of bending angle-based tropopauses to lapse rate tropopauses latitudinally resolved, and included comparisons to collocated radiosondes. Schmidt et al. 2010 used 8 years of RO data from various missions and ECMWF analysis data and showed latitudinally and seasonally resolved differences between bending angle- and dry temperature-based tropopauses. They also compared differences in tropopause height trends between these methods.

Apart from the comparison between FY3C and Metop (see above), I am not sure how the new findings of this study in light of the existing ones can be summarized. I think

the main results and conclusions of this study need to be discussed in more detail, going beyond the existing results. If the main findings are the differences between Metop and FY3C, then the different processing needs to be addressed (see above), and some attempt to explain those differences need to be made. If the focus stays on the difference between LRT and BAT, the details of Figs. 11 and 12 are worth to be looked at in more detail and additional work should be performed to understand these differences better (with the existing work as background).

** Specific comments

* The global map plots (Figs. 3, 5, 7, 8) are hard to read for several reasons. For the difference plots (bottom row), please use a diverging color scale so that one can distinguish positive and negative values. Secondly, the resolution of the bitmaps is too low. And thirdly, I do not understand where the white stripes pattern in these plots comes from? They do not represent the RO event distribution, and seem to be related to the underlying map. I find these patterns quite distracting.

* The paper needs heavy editing with respect to English language. I am (obviously) not a native speaker myself, and strongly recommend to use an English proofreading service.

* You write that you use "ECMWF reanalysis data"; I guess you actually use ECMWF operational analysis data?

* In Fig. 6, and also Fig. 4, the presented bias structures of Metop and FY3C are very different. Do you have an explanation for this? I think this is quite surprising, and I would guess that this is related to the different data processings (see above). Largest differences can be found in mid latitudes, where the occurrence of double tropopauses make the detection of the first tropopause more demanding. If the paper keeps the comparison between the two missions, these differences need to be discussed and analyzed in more detail (e.g. single profile comparisons, and so on). As shown in a number of publications, RO measurements from different missions can be usually com-
bined as long as they are processed consistently, such systematic differences between two missions are therefore surprising to the reader and need explanation.

** Minor comments and technical corrections

* p.1 l30: "variation trend"? Probably you mean "trend in tropopause height"?

* p.2 l33: "direct sounding technic" => "in-situ measurements"

* p.3 l65: "Bending angle profile is level-1 data but dry temperature profile is level-2 data, and thus we only discuss the tropopause height (TPH)" I do not understand this sentence.

* Fig. 1: "Temperature" is misspelled.

* p.6 l117: GPS, BDS: Introduce acronyms

* p.6, l121: "Wegner" => "Wegener"

* p.6, l124: Which processing is used?

* p.7 l141-142: Supposing that these are ECMWF operational analysis, are these numbers correct? To my knowledge analysis resolution close to the surface is better than the stated 200 m, and at 20 km it should be around 400 m (and not 300 m), but I might be wrong.

* p.7 l144: "bias regulation": Not sure what this means. Maybe "bias characteristics"? This term occurs several times in the manuscript.

* p.7 l149: "spatial interval with" => "spatial distance to"

* p.8 l169: "opposite trend" => "opposite behavior" ("trend" is misleading).

* p.9 l176: "totally different": This is overstated, I think they look overall quite similar.

* p.12 l215: "bias trends": No trends involved here.

* p.12 l216: "RMSE trend": No trends involved here.

**AMTD**

* p.13 l238: "variation trends": No trends involved here.

* Figs. 11, 12: "Summer" and "Winter" are ambiguous. Please use "DJF" and "JJA".

* p.15 l252: "To detect the global tropopause" => "To detect the global tropopause with RO"

* p.16 l262: No trend involved here.

* p.16 l262: "availability" is not the correct word. Maybe "reliability" or "usability"?
* * *

---

## Author Comment (AC4) · 17 Feb 2020

Dear Anonymous Referee 2 and editor Thanks for reading our manuscript and providing comments and suggestions, which are really helpful for us. We have carefully read your comments and gave the following responses. Massive revisions are done, and thus we upload the revised manuscript in supplement file. We also uploaded this reply in PDF format in supplement file.

For general comments: 1. * Using RO data from two missions introduces a complication to this study. I see two issues here: First, I think it would help the paper to keep its focus on the comparison of the two tropopause determination methods, instead of also comparing two missions. Secondly, and more importantly, the data of these missions

are from two different processing centers and I suppose that they are not consistently processed. Metop data are processed by ROMSAF, and information about the processing of FY3C is missing in the paper. It has been shown in the literature that only for the same processor and processing version profiles from different RO missions can be mixed together. Applying the tropopause determination algorithms on inconsistently processed profiles will most probably result in a systematic bias. I would strongly recommend to only use data from one processing to avoid this additional complication, or to first validate that no bias is introduced.

Our reply: Thanks for your advice. Using only one RO mission's data can avoid systematic bias, which is better for comparing two methods. However, the data profile obtained by GNSS RO is globally random distributed, and now, and only few RO missions are capable for long term stable operational running, mainly including FY3, Metop, COSMIC (equipment aging) and COSMIC2 (launched recently), and the amount of their daily data profile is still limited. To study tropopause and other global scale atmospheric researches based on GNSS RO data, the best way is to assimilate all RO missions' data. Thus, studying the consistency between different RO missions' data and the bias characteristic of different RO missions' data is the first step to achieve the inter-mission data assimilation. Our article focuses on the tropopause, and shows the bias characteristic of tropopause derived from different RO mission's data (FY3C/Metop), and it is the major objective of our article. BesidesïijŇthe other purpose of this article is to promote FY3 atmosphere products. FY3C has already be in orbit for 6 years, and data from FY3C is prepared to be used in long term atmospheric research. Thus, bias comparison between FY3C and Metop as well as ECMWF operational data is quite meaningful for the participation of FY3C in RO international cooperation in the future and also helpful for the further quality improvement of FY3 products. Based on this reason, we used two RO missions in our study. In spite of this, in our results, we provided comparison results of two methods based on only one RO mission data (ie. Fig. 11 and 12 in manuscript). Also, for the bias between RO and ECMWF, if you are only interested in seasonal bias of Metop, you can only focus on the solid line in Fig. 4 and 6 in manuscript, as well as

Fig. 9 and 10 in manuscript. If we add figures for seasonal bias between single RO mission and ECMWF, the results may seem repetitive. The systematic error caused by different processor will affect the comparison results. Data of FY3 is processed by China Meteorological Administration (we add this in line 27 in revised manuscript) and the part of processing software (from Level 1b (bending angle/impact parameter/orbit parameters...etc) to Level 2 (temperature/pressure/humidity...etc)) of FY3 is based on ROPP, which is similar with Metop (line 18), besides, the local solar time of FY3C and Metop is similar (line 18), and it is the reason why we choose FY3C and Metop in our study. ROMSAF website provide local solar time and global statistics of bending angle/refractivity of many RO missions, including FY3C and Metop. The following figures give local solar time comparison and the one-day global statistic for bending angle from Metop-a, Metop-b and FY3C respectively (we just provide an example, you can find it on https://www.romsaf.org/monitoring/index.php).

(There are some problems about uploading figures via this system, and thus we upload these 5 figures in the supplement file. However, we suggest you reading the pdf format reply in supplement file directly.)

Those results are quite same. However, global statistic may cannot reflect the results in some certain area, and it is impossible to show global distribution of bending angle/refractive/temperature, and thus in our RO tropopause results, some regional biases are shown and it is meaningful for the validation and the further development of FY3.

2. *Lewis 2009 showed comparisons of bending angle-based tropopauses to lapse rate tropopauses latitudinally resolved, and included comparisons to collocated radiosondes. Schmidt et al. 2010 used 8 years of RO data from various missions and ECMWF analysis data and showed latitudinally and seasonally resolved differences between bending angle- and dry temperature-based tropopauses. They also compared differences in tropopause height trends between these methods. Apart from the comparison between FY3C and Metop (see above), I am not sure how the new finding of this study

in light of the existing ones can be summarized. I think the main results and conclusions of this study need to be discussed in more detail, going beyond the existing results. If the main finding are the differences between Metop and FY3C, then the different processing needs to be addressed (see above), and some attempt to explain those differences need to be made. If the focus stays on the difference between LRT and BAT, the details of Figs. 11 and 12 are worth to be looked at in more detail and additional work should be performed to understand these differences better (with the existing work as background). Our response: RO tropopause is not a new topic, which has been studied since the launch of GPS-MET experiment. For the tropopause results, we just provide these comparison results derived from FY3C and Metop to show the bias characteristic for others who use these data or de relative research to take reference. Also, the part of our purpose, as we noted above, is to promote and improve FY3 series data. For example, although the global statistic of bending angle of two RO missions are similar, FY3C shows higher BAT than Metop, especially over Antarctic during DJF and over 45N during JJA. The LRT of two RO missions show different over mid latitude region. These point out the problem that need to be investigated. However, these are more relative to the data processing instead of tropopause. The FY3C data is not completely processed by us and the inner process of retrieval from RO raw data to temperature is complex, and thus we cannot answer the reason of these difference exactly, now. This need further study.

For specific comments: 1. * The global map plots (Figs. 3, 5, 7, 8) are hard to read for several reasons. For the difference plots (bottom row), please use a diverging color scale so that one can distinguish positive and negative values. Secondly, the resolution of the bitmaps is too low. And thirdly, I do not understand where the white stripes pattern in these plots comes from? They do not represent the RO event distribution, and seem to be related to the underlying map. I find these patterns quite distracting. Our response: We have renewed the figures in revised manuscript. Each color point represents a tropopause height derived from a bending angle/atmosphere profile. The white stripes indicate that there is no data in this area. These figures illustrate the data

distribution, indicating that data of FY3C and Metop has the similar distribution.

2. * The paper needs heavy editing with respect to English language. I am (obviously) not a native speaker myself, and strongly recommend to use an English proofreading service. Our response: We have asked language edit service to polish our manuscript. The revised manuscript and the language edit certificate is uploaded in supplement file.

3. * You write that you use "ECMWF reanalysis data"; I guess you actually use ECMWF operational analysis data? Our response: We made a mistake here, and we have corrected it as "operational analysis data".

4. * In Fig. 6, and also Fig. 4, the presented bias structures of Metop and FY3C are very different. Do you have an explanation for this? I think this is quite surprising, and I would guess that this is related to the different data processings (see above). Largest differences can be found in mid latitudes, where the occurrence of double tropopauses make the detection of the first tropopause more demanding. If the paper keeps the comparison between the two missions, these differences need to be discussed and analyzed in more detail (e.g. single profile comparisons, and so on). As shown in a number of publications, RO measurements from different missions can be usually combined as long as they are processed consistently, such systematic differences between two missions are therefore surprising to the reader and need explanation.

Our response: Although multi-tropopause usually occurs in mid-latitude, we do not think the different bias in Fig. 4 and 6 is caused by multi-tropopause. The reasons are as follow: LRT is determined according to WMO criterion (scan tropopause from bottom to top). The multi-tropopause may cause the first tropopause not obvious, but it is still data problem that one always find the first tropopause while the other always find the second tropopause (if so). On the contrary, in Fig. 11a and 12a, the BAT is higher than LRT over tropics and polar region, but they are similar over mid-latitude. we think it may be caused by multi-tropopause, for BAT is to find the strongest tropopause, but

LRT is to find the lowest tropopause. The difference in Fig. 4 and 6 are more likely caused by the data itself, in other words, the inner processing of data. As we noted above, FY3C and Metop have the same local time and the same processing software. We try to avoid systematic bias and thus we choose FY3C and Metop, but the results are still different. Thus, to find out the answer, it needs to look over each step of data retrieval. It needs further study, and we cannot give the answer right now in this article. For the data combination, actually we did not combine data from two RO missions. Results of FY3C is just based FY3C data and results of Metop is just based on Metop data.

5. * p.1 l30: "variation trend"? Probably you mean "trend in tropopause height"? Our response: We use 'changes in the tropopause' in revised manuscript.

6. * p.2 l33: "direct sounding technic" => "in-situ measurements" Our response: We corrected this according to your suggestion.

7. * p.3 l65: "Bending angle profile is level-1 data but dry temperature profile is level-2 data, and thus we only discuss the tropopause height (TPH)" I do not understand this sentence. Our response: We noticed that level-1, level-2 is not an official terminology, which are just used in RO retrieval, and thus we delete this sentence.

8. * Fig. 1: "Temperature" is misspelled. Our response: We have corrected this mistake.

9. * p.6 l117: GPS, BDS: Introduce acronyms Our response: We add the full name of the acronyms.

10. * p.6, l121: "Wegner" => "Wegener" Our response: We have corrected this mistake.

11. * p.6, l124: Which processing is used? Our response: We add "The data processing of FY3 is undertaken by China Meteorological Administration (CMA)". The processing software is also ROPP, which is noted in line 118.

12. *p.7l141-142: Supposing that these are ECMWF operational analysis, are these

numbers correct? To my knowledge analysis resolution close to the surface is better than the stated 200 m, and at 20 km it should be around 400 m (and not 300 m), but I might be wrong.

Our response: Yes, it is ECMWF operational analysis, and we have corrected this in revised manuscript. The first level is 0.02 km and the second level is 0.04 km. Until 2km, the interval is 200 m. The 90th level is 19.83 km and 91th level is 20.22 km. At 17 km, the interval is about 300m, and tropopause seldom higher than 17 km. From 200 m at 0 km to 300 m at 20 km is not accurate, but if we say 20m-400m, it may still lead misunderstanding because the interval variation is not homogenous. We corrected as "from 200 m at 2 km to 300 m at 17 km (tropopause is seldom higher than 17 km) and 400 m at 20 km."

13. * p.7 l144: "bias regulation": Not sure what this means. Maybe "bias characteristics"? This term occurs several times in the manuscript. Our response: We correct it as "latitudinal bias". We also correct this term in later section.

14. * p.7 l149: "spatial interval with" => "spatial distance to" Our response: We corrected this according to your suggestion.

15. * p.8 l169: "opposite trend" => "opposite behavior" ("trend" is misleading). Our response: We corrected this according to your suggestion.

16. * p.9 l176: "totally different": This is overstated, I think they look overall quite similar. Our response: There indeed has some difference, and we delete the "totally".

17. * p.12 l215: "bias trends": No trends involved here. Our response: We delete the "trends" here.

18. * p.12 l216: "RMSE trend": No trends involved here. Our response: We use "curve", instead of "trend".

19. * p.13 l238: "variation trends": No trends involved here. Our response: We delete the "trends" here.

20. * Figs. 11, 12: "Summer" and "Winter" are ambiguous. Please use "DJF" and "JJA". Our response: We have renewed the figures

21. * p.15 l252: "To detect the global tropopause" => "To detect the global tropopause with RO" Our response: We corrected this according to your suggestion.

22. * p.16 l262: No trend involved here. Our response: We delete the "trends" here.

23. * p.16 l262: "availability" is not the correct word. Maybe "reliability" or "usability"? Our response: We use reliability here.

If you have any question, please do not hesitate to contact us. We are hoping for hearing from you.

Please also note the supplement to this comment:
https://www.atmos-meas-tech-discuss.net/amt-2019-379/amt-2019-379-AC4-supplement.zip

---

## Author Comment (AC5) · 1 Mar 2020

Thanks for your earnest work on reviewing our manuscript. We missed this comment before, and now we are requested to respond to all referee comments. We are sorry if this comment have bothered you.